# JAX-SPH: A DIFFERENTIABLE SMOOTHED PARTICLE HYDRODYNAMICS FRAMEWORK

**Artur P. Toshev**[†,1]**, Harish Ramachandran**[*,1]**, Jonas A. Erbesdobler**[*,1]**, Gianluca Galletti** [*,2]
**Johannes Brandstetter**[3,4] **& Nikolaus A. Adams**[1,5]

[1] Chair of Aerodynamics and Fluid Mechanics, TUM, Germany
[2] Independent researcher
[3] ELLIS Unit Linz, LIT AI Lab, Institute for Machine Learning, JKU Linz, Austria
[4] NXAI GmbH, Linz, Austria
[5] Munich Institute of Integrated Materials, Energy and Process Engineering, TUM, Germany
[†] `artur.toshev@tum.de`
[*] Equal contribution

## ABSTRACT

Particle-based fluid simulations have emerged as a powerful tool for solving the Navier-Stokes equations, especially in cases that include intricate physics and free surfaces. The recent addition of machine learning methods to the toolbox for solving such problems is pushing the boundary of the quality vs. speed trade-off of such numerical simulations. In this work, we lead the way to Lagrangian fluid simulators compatible with deep learning frameworks, and propose JAX-SPH – a Smoothed Particle Hydrodynamics (SPH) framework implemented in JAX. JAX-SPH builds on the code for dataset generation from the LagrangeBench project (Toshev et al., 2024a) and extends this code in multiple ways: (a) integration of further key SPH algorithms, (b) restructuring the code toward a Python package, (c) verification of the gradients through the solver, and (d) demonstration of the utility of the gradients for solving inverse problems as well as a Solver-in-the-Loop application. Our code is available at https://github.com/tumaer/jax-sph.

## 1 INTRODUCTION

Partial differential equations (PDEs) are the mathematical tools developed to describe natural phenomena ranging from engineering and physics to social sciences. Various numerical methods have been developed to solve these PDEs, as analytical solutions are only available for toy examples, with the most recent class of PDE solvers being the machine learning-based ones (Thuerey et al., 2021; Brunton & Kutz, 2023). One particular class of machine learning (ML) approaches called hybrid solvers refers to combining ideas (or full algorithmic blocks) from classical numerical solvers and machine learning (Schenck & Fox, 2018; Um et al., 2020; Kochkov et al., 2021; Jagtap et al., 2022; Lienen & Günnemann, 2022; Karlbauer et al., 2022; Li & Farimani, 2022; Toshev et al., 2023b).

This has been one of the main reasons for the development of differentiable fluid mechanics solvers like PhiFlow (Holl et al., 2020), JAX-CFD (Kochkov et al., 2021), and JAX-Fluids (Bezgin et al., 2022). However, these three frameworks implement Eulerian, i.e., grid-based, solvers, and we lead the way to a JAX-based Lagrangian CFD solver. Eulerian solvers refer to numerical methods that discretize space into static volume elements and then track the evolution of the fluid properties at these positions, while Lagrangian solvers discretize individual material elements, which are then shifted in space following the local velocity field.

Algorithmically, grid-based and particle-based methods are very different. While grid-based solvers rely on stencils akin to kernels in Convolutional Neural Networks (CNNs) (Lecun et al., 1998), particle-based solvers rely on kernel approximations akin to Graph Neural Networks (GNNs) (Scarselli et al., 2008; Battaglia et al., 2018) operating on a radial distance-based graph. The main overhead of Lagrangian over Eulerian approaches is updating the connectivity between discretization elements at every autoregressive solver step. Even if an Eulerian scheme operates on

an irregular mesh, the connectivity prescribed by this mesh can be precomputed, while by moving particles in space in Lagrangian methods, their neighbors constantly vary in time.

As SPH techniques advance, various software packages have arisen in the past few years. However, most of them are designed for high-performance computing (HPC) applications and are typically implemented in low-level languages like C++ or Fortran (Crespo et al., 2015; Koschier et al., 2019; Zhang et al., 2021), with two notable exceptions: PySPH (Ramachandran et al., 2021) in Python and juSPH (Luo et al., 2022) in Julia. Nevertheless, hardly an SPH implementation exists which is out-of-the-box compatible with modern deep learning frameworks like TensorFlow (Abadi et al., 2015), PyTorch[1] (Paszke et al., 2019) or JAX (Bradbury et al., 2018), i.e., leverages automatic differentiation to power differentiable solvers like PhiFlow (Holl et al., 2020). Upon identifying the lack of an ML-ready SPH solver, we choose JAX for the framework implementation for two reasons: (a) JAX tends to be faster for operations on graphs even after PyTorch 2.0 has been introduced (see Appendix F in Toshev et al. (2024a)), and (b) we can use the cell list-based neighbor search routine of the JAX-MD library (Schoenholz & Cubuk, 2020). We note that with our solver, we target the easier integration of SPH with ML workflows rather than developing a parallel HPC code, and we exclusively use the AD routine `grad` by JAX. We leave the implementation of better custom adjoints along the lines of Ma et al. (2021); Kidger (2022); Nadarajah & Jameson (2000) to future work.

In this work, we significantly extend the codebase used for dataset generation within LagrangeBench[2] (Toshev et al., 2024a) and demonstrate the utility of the obtained gradients in multiple ways. Our contributions are:

- The addition and validation of Transport Velocity (Adami et al., 2013), Riemann SPH (Zhang et al., 2017b), and thermal diffusion effects (Cleary, 1998).
- Validating the accuracy of the automatic differentiation-based gradients (Griewank & Walther, 2008) over 5 solver steps.
- An open-source Python package at pypi.org/project/jax-sph.
- A demonstration of how to use these gradients on (a) an inverse problem over 100 SPH solver steps, and (b) using the SPH solver in a Solver-in-the-Loop fashion (Um et al., 2020).

## 2 SPH SOLVER

In this section, we introduce the components included in our solver code. The validation of these can be found in Appendix A, for which we include various solver validation cases.

**Weakly compressible SPH.** We follow the weakly compressible SPH approach (Monaghan, 1994; Morris et al., 1997) to evolve the dynamics of incompressible fluids. The equations governing such systems are the mass and momentum conservation equations

$$\frac{\mathrm{d}}{\mathrm{d}t}(\rho) = -\rho \left(\nabla \cdot \mathbf{u}\right),$$  (1)

$$\frac{\mathrm{d}}{\mathrm{d}t}(\mathbf{u}) = \underbrace{-\frac{1}{\rho}\nabla p}_{\text{pressure}} + \underbrace{\frac{1}{Re}\nabla^2\mathbf{u}}_{\text{viscosity}} + \underbrace{\mathbf{f}}_{\text{ext. force}},$$  (2)

with density $\rho$, velocity $\mathbf{u}$, pressure $p$, Reynolds number $Re$, and external force $\mathbf{f}$. To numerically solve these equations, SPH applies a distance-based kernel $W$ that averages over the properties of the fluid. The default kernel in our codebase is the Quintic spline (Morris et al., 1997), but we also include the 5th order Wendland kernel (Wendland, 1995).

The preferred way of estimating the density (Monaghan, 2005) is through *density summation* $\rho_i = \sum_j m_j W(r_{ij}|h)$, with $m_j$ being the mass of the adjacent particles $j$, $h$ the smoothing length of the kernel, and $r_{ij}$ the interparticle distance. However, this approach leads to unphysically low

---

[1]For similar work in progress, we refer the reader to TorchSPH https://github.com/wi-re/pytorchSPH. Compared to our JAX-SPH, this code uses PyTorch, different SPH algorithms, different experiments, and does not validate the solver or the gradients.

[2]https://github.com/tumaer/lagrangebench/blob/main/notebooks/data_gen.ipynb.

densities at free surfaces, forcing the use of *density evolution*, i.e., numerically integrating the mass conservation Eq. 1. We also implement a density field reinitialization method (Zhang et al., 2017a) to mitigate errors arising from density evolution (Colagrossi & Landrini, 2003). In weakly compressible SPH, the pressure is defined as a function of density through an equation of state, and in our code, we choose the formulation by Monaghan (1994) for standard and transport velocity formulation SPH. The equation of state used for the Riemann SPH formulation is the one proposed by Zhang et al. (2017b).

**Transport velocity.**   The observation that tensile instabilities in standard SPH can cause particle clumping and void regions (Price, 2012) has led to the development of advanced shifting schemes like the transport velocity formulation of SPH (Adami et al., 2013; Zhang et al., 2017a). We add the shifting velocity proposed by Adami et al. (2013) as an optional feature in our codebase.

**Riemann SPH.**   We also add the Riemann SPH solver formulation by Zhang et al. (2017b), which is based on the idea of introducing a simple low dissipation limiter to a classical Riemann solver, resulting in decreased numerical dissipation. We construct a one-dimensional linear inter-particle Riemann problem along a unit vector pointing from particle $i$ to particle $j$, which implicitly regularizes both the momentum equation and the mass conservation. We implement the dissipation limiter as proposed by Zhang et al. (2017b).

**Wall boundaries.**   We follow the generalized wall boundary condition approach by Adami et al. (2012) to enforce different boundary conditions for the standard SPH and the transport velocity SPH formulation. This approach, in essence, implements two physical constraints: (a) impermeability of walls and (b) viscous effects at walls. For impermeability, the pressure of the adjacent fluid particles is assigned to the dummy wall particles, which leads to a zero pressure gradient in the wall-normal direction at the wall surface, and, thus prevents the penetration of fluid particles into the wall. Regarding viscosity, there are two cases to distinguish: (b1) *no-slip* enforces the no-slip boundary condition, i.e., the fluid must have zero velocity directly at the wall surface, and (b2) *free-slip*, i.e., the fluid must have a zero velocity normal to a wall, but might have any velocity tangentially. These two conditions are implemented by assigning the inverted velocity of the fluid to the wall particles, either fully for (b1) or only in the wall-normal direction for (b2). See Adami et al. (2012) for more details.

For the Riemann SPH wall boundary implementation, we solve a one-sided Riemann problem as proposed by Zhang et al. (2017b). Unlike the generalized wall boundary condition, this method avoids interpolating and then extrapolating the fluid states for the wall boundary particles. Here, these wall particles are assigned the individual Riemann states depending on the current particle-particle interaction. For a more in-depth explanation, see (Zhang et al., 2017b; Yang et al., 2020).

**Thermal diffusion.**   Thermal diffusion in weakly compressible SPH involves the transfer of heat between neighboring particles governed by Fourier's law of heat (Cleary, 1998). This diffusion smoothens the temperature field by applying an SPH kernel interpolation over adjacent particles to compute the rate of temperature change. As we deal with incompressible fluids, the dynamics are not influenced by thermal effects, and the pressure or velocity fields do not directly influence the temperature field. Thus, the addition of temperature does not interfere with the previously presented SPH algorithms and allows us to study thermal effects which are governed by diffusion (by our explicit diffusion modeling) and convection (as Lagrangian particles are shifted in space). For an example of a channel flow with a hot wall, see Appendix C.

## 3   EXPERIMENTS

**Gradient validation.**   To test the validity of the gradients obtained through our differentiable solver, we compute analytical solver gradients via automatic differentiation and compare those to numerical gradients from finite difference schemes (Griewank & Walther, 2008). In our setup, gradients are accumulated over 5 solver steps, preceded by 10 (forward only) warm-up steps. Epsilon for finite differences is picked as $0.001dx = 5e - 5$, as smaller values lead to instabilities. Fig. 1 shows the scalar gradients of kinetic energy over position changes $\frac{dE_{kin}}{dr}$ when using 2-dimensional Taylor-Green vortex (TGV) (Brachet et al., 1983; 1984) and lid-driven cavity (LDC) (Ghia et al., 1982) as initial states.

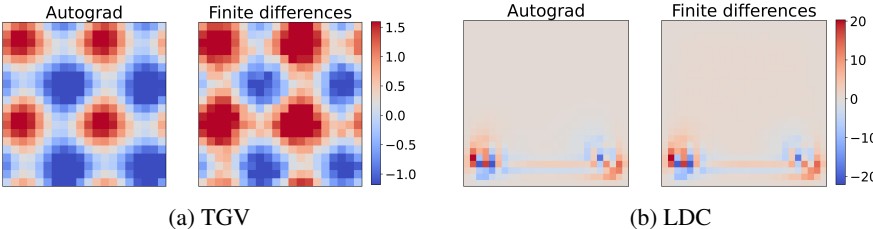

Figure 1: Gradient magnitudes with JAX Autograd and with finite differences on Taylor-Green vortex (left) and lid-driven cavity (right).

**Inverse problem.** Our first application case is an inverse problem, representing the class of inverse design and flow control problems that are successfully tackled by differentiable solvers (Holl et al., 2020) as well as differentiable learned solver surrogates (Allen et al., 2022). The scenario involves a 2D box containing a water cube, discretized by 36 particles, which undergoes acceleration due to gravity across 100 solver steps. The task is to find the initial coordinates of the cube given its final state; see Fig. 2.

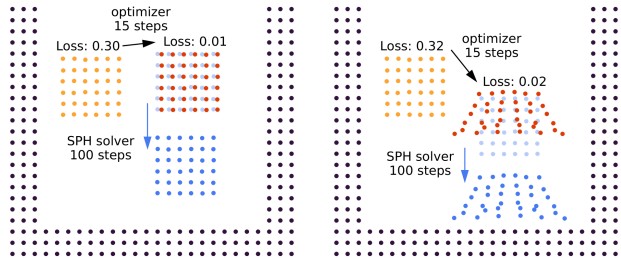

Figure 2: Inverse problems of finding the initial coordinates (light blue) given the final coordinates (blue) of a falling water cube simulation spanning 100 SPH steps. The optimization spans 15 gradient descent steps from orange to red. Free fall case (left) and wall-interactions (right).

The inverse problem is formulated by computing the mean squared error of coordinates between the target final state and the end of a simulation with randomly placed initial particles. After as few as 15 gradient descent steps, we reach a state closely resembling the original one, up to some loss of information during the deformation of the water cube while interacting with the wall.

**Solver-in-the-loop.** As a second experiment to showcase the solver differentiability, we adapt the popular *"Solver-in-the-Loop"* (SitL) (Um et al., 2020) training scheme to SPH. Initially developed to tackle spatial coarsening on grids, SitL interleaves a solver operating on a coarse spatial and/or temporal discretization with a learnable correction function. The solver manages coarse, low-frequency components, while the learnable function adjusts high-frequency details. Due to the inherent difficulties of spatial coarse-graining in particle systems, our objective is to implement SitL only for temporal coarsening. This significantly differs from the original application of SitL and mandates a series of design changes to the original architecture, mainly related to the normalization and training procedure, see Appendix B.

Fig. 3 shows the time evolution of a 2D reverse Poiseuille flow (RPF) (Fedosov et al., 2008) dataset similar to the one in Toshev et al. (2024a) but consisting of positions sampled every 20th ground truth SPH step. Fig. 3 compares only employing the coarse SPH solver with $L = 3$ intermediate steps (left), a fully learned GNS model (Sanchez-Gonzalez et al., 2020) without intermediate steps (middle), and SitL using the same GNS model, but having three intermediate steps of GNS and SPH (right). More details on the training and quantitative results can be found in Appendix B.2.

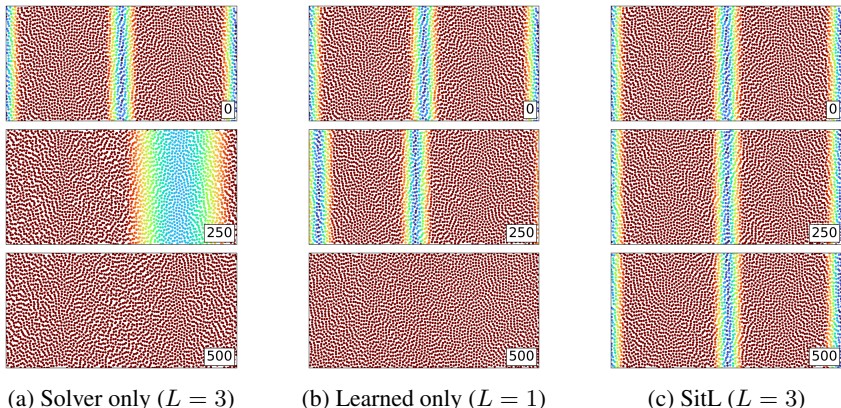

(a) Solver only ($L = 3$)  (b) Learned only ($L = 1$)  (c) SitL ($L = 3$)

Figure 3: Evolution of the velocity magnitude in reverse Poiseuille flow. The stamps in each image refer to the respective step of the original SPH simulation, i.e., 500 means 10 000 SPH steps.

## 4 CONCLUSION

We have developed JAX-SPH, a framework for simulating Lagrangian fluid problems, which can be easily integrated into design/control problems as well as hybrid solver approaches. By building our code on the high-performance library JAX and validating the simulation results of our solver, we offer a fast and reliable SPH solver in Python. With our work, we hope to accelerate the development of more hybrid Lagrangian solvers, e.g., Toshev et al. (2024b), and we leave the addition of more SPH algorithms and simulation cases to future work. One particularly exciting future direction is developing foundation models for PDEs that can operate on both Eulerian and Lagrangian data (Alkin et al., 2024), potentially combined with encoded symmetries (Toshev et al., 2023a).

## ACKNOWLEDGEMENTS

The authors thank Fabian Fritz for providing an initial JAX implementation of the 3D Taylor-Green vortex simulated with the transport velocity SPH formulation by Adami et al. (2013). The authors also thank Xiangyu Hu and Christopher Zöller for helpful discussions on SPH, and Ludger Paehler for discussions on gradient validation.

## AUTHOR CONTRIBUTIONS

A.T. developed the codebase, selected the SPH algorithms and validated most of them, designed the simulation cases and experiments, ran many of them, and wrote the first version of the manuscript. H.R. implemented thermal diffusion and the inverse problem, and helped in the initial phase of Solver-in-the-Loop. J.E. implemented Riemann SPH, generated the final solver validation results, and contributed to refactoring the core SPH code. G.G. validated the gradients through the solver, implemented Solver-in-the-Loop, and tuned its parameters on the dataset A.T. provided. J.B. contributed to the choice of experiments showcasing the gradients through the solver. N.A. supervised the project from conception to design of experiments and analysis of the results. All authors contributed to the manuscript.

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

## A  SOLVER VALIDATION

### A.1  TAYLOR GREEN VORTEX 2D

Fig. 4 shows the absolute velocities of each particle at the start and at the end of the Taylor Green vortex (Brachet et al., 1983; 1984) simulation at $Re = 100$. Here, the transport velocity SPH formulation is used. In Fig. 5, one can see the decay of the maximum velocity and the kinetic energy over time for the different methods, standard SPH (SPH), transport velocity formulation SPH (SPH + tvf), and Riemann SPH (Riemann).

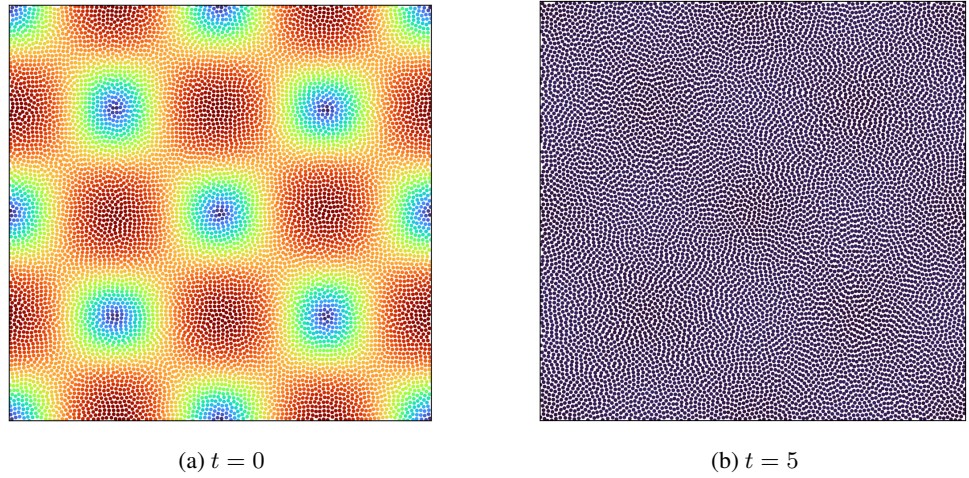

(a) $t = 0$                                    (b) $t = 5$

Figure 4: 2D Taylor Green vortex velocity magnitudes at the start of the simulation (left) and at $t = 5$ (right), calculated using transport velocity formulation SPH.

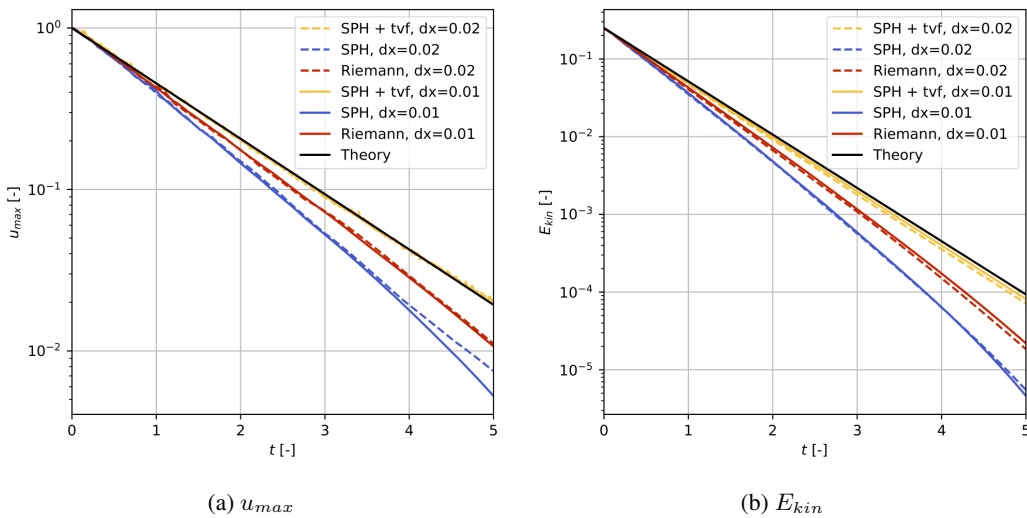

(a) $u_{max}$                                   (b) $E_{kin}$

Figure 5: 2D Taylor Green vortex SPH method comparison for $u_{max}$ (left) and $E_{kin}$ (right) between standard SPH, transport velocity formulation SPH, and Riemann SPH at $dx = 0.02$ and $dx = 0.01$.

### A.2  LID-DRIVEN CAVITY 2D

The following figures compare the different methods for a 2D lid-driven cavity (Ghia et al., 1982) simulation at $Re = 100$. The reference data for the velocity profiles, i.e., black dots for U (velocity

in $x$ direction at a vertical cut through the middle of the cavity) and black squares for V (velocity in $y$ direction at a horizontal cut through the middle of the cavity), are from Ghia et al. (1982).

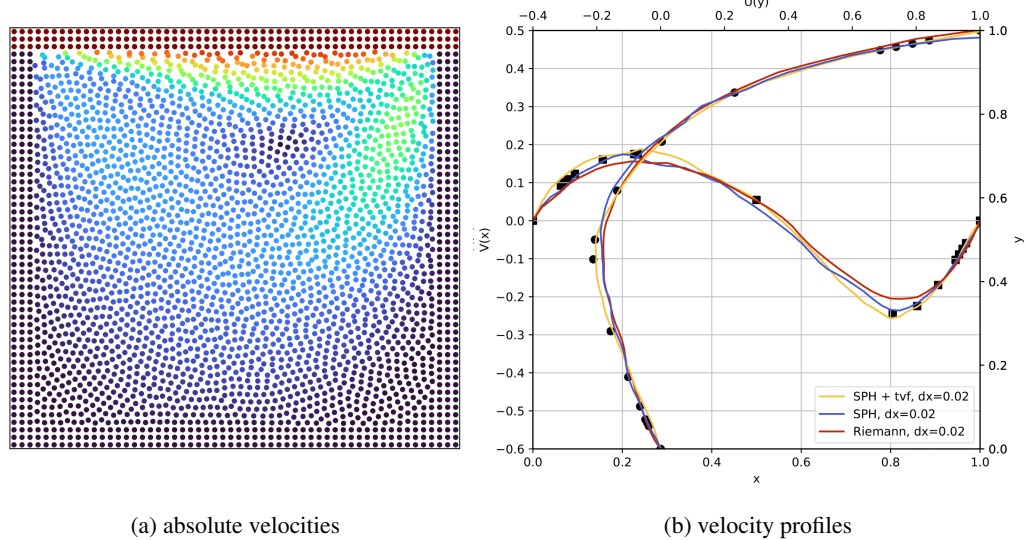

(a) absolute velocities                    (b) velocity profiles

Figure 6: Lid-driven cavity with $dx = 0.02$ showing absolute particle velocities of the Riemann solver (left) and velocity profiles of each SPH method at the midsection for U and V (right)

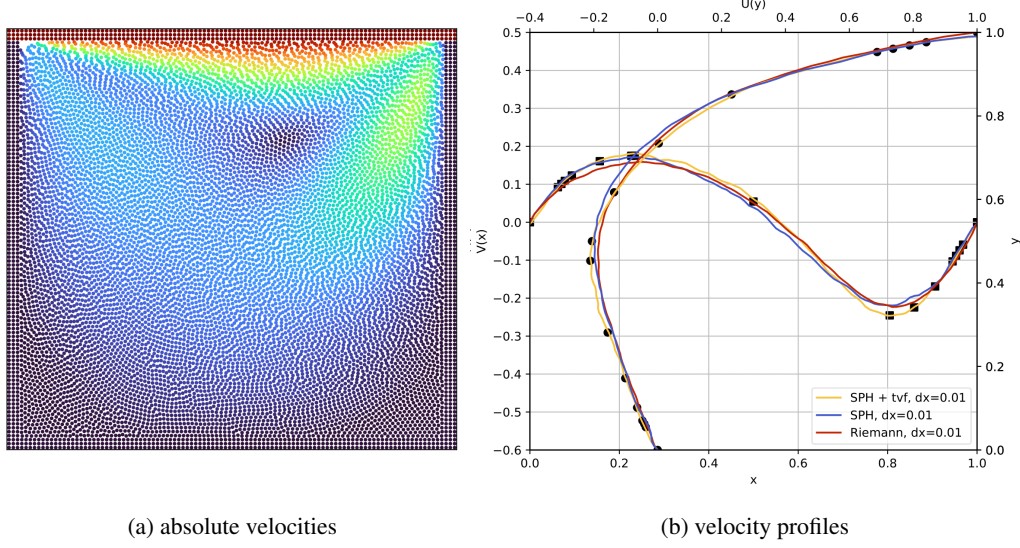

(a) absolute velocities                    (b) velocity profiles

Figure 7: Lid-driven cavity with $dx = 0.01$ showing absolute particle velocities of the Riemann solver (left) and velocity profiles of each SPH method at the midsection for U and V (right)

## A.3 DAM BREAK 2D

The following Fig. 8 shows the nondimensionalized pressure of a Riemann SPH (Zhang et al., 2017b) dam break (Colagrossi & Landrini, 2003) simulation at different time stamps. The fluid flows from the initial state on the left to the right side, interacts with the wall, and reflects a wave backward throughout the domain. A similar validation plot for the standard SPH formulation of our solver is presented in Appendix B of Toshev et al. (2024a).

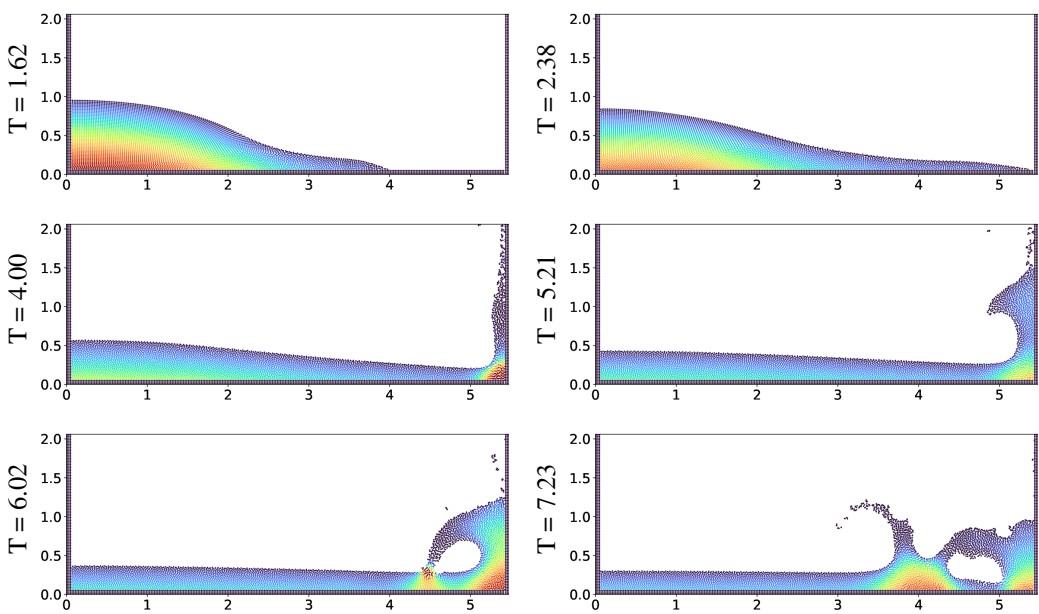

Figure 8: Dam break simulation with $dx = 0.02$ using Riemann SPH at different time stamps T, visualizing the nondimensionalized pressure

### A.4 TAYLOR-GREEN VORTEX 3D

Fig. 9 shows the comparison between the different SPH methods on the 3D TGV, similar to Fig. 5 for 2D. Again, the Reynolds number is set to $Re = 100$, and the number of particles in the unit cube is $20^3$, $32^3$, and $50^3$, leading to $dx = 0.314$, $dx = 0.196$, and $dx = 0.126$, respectively. The reference solution is obtained using JAX-Fluids Bezgin et al. (2022) with a $128^3$ grid.

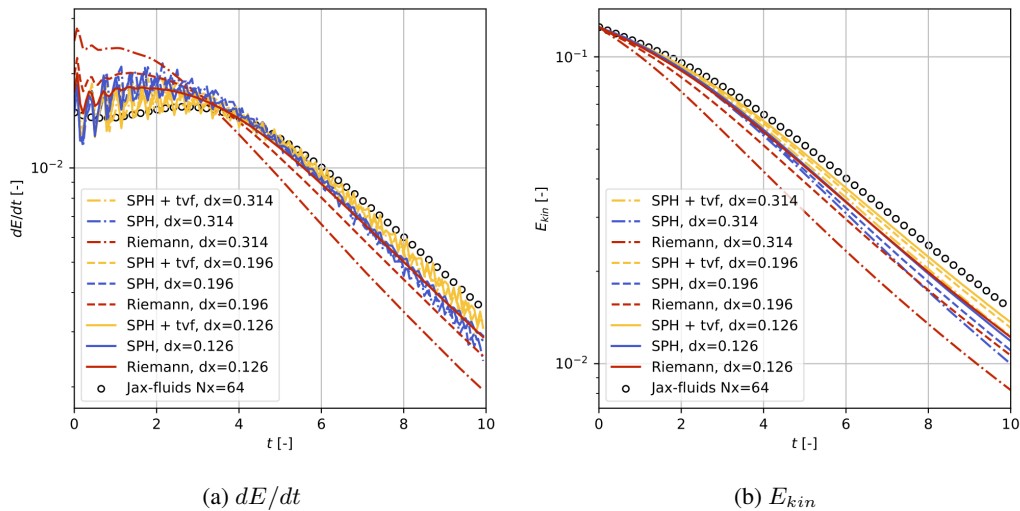

(a) $dE/dt$               (b) $E_{kin}$

Figure 9: 3D Taylor Green vortex comparison for $dE/dt$ (left) and $E_{kin}$ (right) between standard SPH, transport velocity formulation SPH, and Riemann SPH with $dx = 0.314$, $dx = 0.196$, and $dx = 0.126$

## B  SOLVER-IN-THE-LOOP

### B.1  IMPLEMENTATION DETAILS

To compute velocities, we apply a finite difference approximation of positions as

$$\mathbf{u}_{(i+1)M} = \mathbf{x}_{(i+1)M} - \mathbf{x}_{iM}, \tag{3}$$

$$\mathbf{a}_{iM} = \mathbf{u}_{(i+1)M} - \mathbf{u}_{iM}. \tag{4}$$

These properties have normalization statistics over the dataset as $\sigma_{uM}$ and $\sigma_{aM}$ (for brevity here, assuming that everything is zero-centered, but we center the data in our code). The values entering and exiting the SitL model should be in normalized space.

Given the number of temporal coarsening steps $M$ and the number of SitL calls n_sitl, each call accounts for $\frac{M}{\text{n\_sitl}}$ actual timesteps. This results in two distinct time steps $(dt)$, necessitating the transformation of input properties to the SitL physical space.

```python
def sitl_forward(self, r, u_M_norm, dt, M, n_sitl, sigma_uM, sigma_aM):
    """Solver-in-the-loop forward call.

    Args:
        r: current coordinates
        u_M_norm: normalized position difference between M SPH steps.
        dt: physical dt from CFL condition
        M: level of temporal coarsening
        n_sitl: number of Solver-in-the-Loop steps
        sigma_uM: std of u_M over the dataset (= std_dx_M)
        sigma_aM: std of a_M over the dataset (= std_ddx_M)
    """
    # transform initial states
    u_phys = u_M_norm * sigma_uM / (dt * M)  # to physical space
    nbrs_GNN = self.nbrs_GNN_update(r)  # GNN neighbor list
    nbrs_SPH = self.nbrs_SPH_update(r)  # SPH neighbor list
    r0, u0, r_new = r.copy(), u_phys.copy(), r

    for l in range(n_sitl):
        # SPH solver call
        a_SPH = self.SPH(u_phys, nbrs_SPH)

        # learned correction
        a_GNN = self.GNN(u_M_norm, nbrs_GNN)
        a_GNN_phys = a_GNN * sigma_aM / (dt * M) ** 2  # to physical

        # add accelerations (in physical space) and integrate
        a_final = a_SPH + a_GNN_phys
        u_phys += (dt * M) / n_sitl * a_final
        r_new += (dt * M) / n_sitl * u_phys

        # update neighbors to new positions
        nbrs_SPH = self.nbrs_SPH_update(r_new)
        nbrs_GNN = self.nbrs_GNN_update(r_new)

        # normalize updated velocity for next GNN input
        u_M_norm = u_phys * (dt * M) / sigma_uM  # to normalized

    # finite difference to get M-step effective quantities
    dx_M = r_new - r0
    dx_M_norm = dx_M / sigma_uM
    ddx_M = dx_M - u0 * (dt * M)
    ddx_M_norm = ddx_M / sigma_aM

    return {"acc": ddx_M_norm, "vel": dx_M_norm}
```
Listing 1: Solver-in-the-Loop forward algorithm

### B.2    TRAINING AND RESULTS

Solver-in-the-Loop was trained with LagrangeBench (Toshev et al., 2024a), with a custom RPF 2D dataset with 20-step temporal coarsening. For SitL, `n_sitl`=3. Both the corrector model and the reference GNS (Sanchez-Gonzalez et al., 2020) are message-passing networks with 10 layers and 64 latent dimensions. The starting learning rate is set to $1e-3$ for both, and `noise_std` is set to $1e-5$ for SitL and to $1e-3$ for GNS. Table 1 shows the LagrangeBench performance metrics on these models. Best models are picked based on the $\text{MSE}_{20}$ loss on the validation set.

| Metric | Solver only | GNS | SitL |
|--------|-------------|-----|------|
| $\text{MSE}_5$ | $1.7e-7$ | $6.7e-9$ | $\mathbf{3.3e-9}$ |
| $\text{MSE}_{20}$ | $7.9e-6$ | $1.9e-7$ | $\mathbf{1.3e-7}$ |
| $\text{MSE}_{E_{kin}}$ | $0.13$ | $2.8e-4$ | $\mathbf{7.4e-5}$ |
| Sinkhorn | $3.4e-7$ | $3.7e-8$ | $\mathbf{9.3e-9}$ |

Table 1: LagrangeBench metrics on the RPF 2D dataset over 20 steps.

## C    THERMAL DIFFUSION EXAMPLE

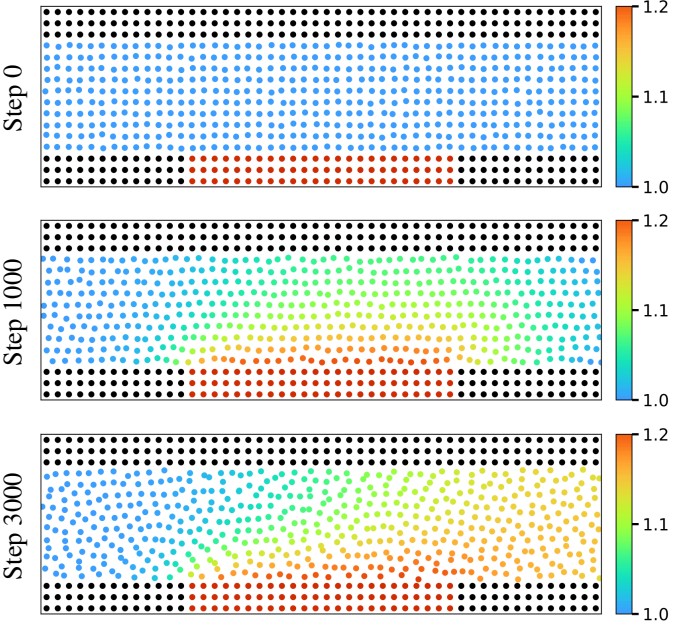

Figure 10: Simulation of channel flow with hot bottom wall using standard SPH and thermal diffusion. The plots show the non-dimensional temperature at different time steps of an SPH simulation.

