# OpenReview forum: "JAX-SPH: A Differentiable Smoothed Particle Hydrodynamics Framework"
_ICLR.cc/2024/Workshop/AI4DiffEqtnsInSci — AI4DiffEqtnsInSci @ ICLR 2024 Poster_

### Official Review · Reviewer_ZrQX · 2024-02-22
**Good contribution to the field of differentiable lagrangian-based solvers**

**Rating:** 7
**Confidence:** 4

**Review:**

**Summary:** The paper introduces a differentiable lagrangian fluid simulator based on smooth particle hydrodynamics (SPH), written in JAX. The authors extend the SPH code in LagrangeBench to be differentiable, demonstrating some applications of the said approach.

**Strengths:**

1. The paper seems well-motivated and addresses the need for differentiable Lagrangian CFD solvers.
2. Extending the differentiability of SPH solvers is a novel contribution with examples demonstrating the usability of the software.
3. Incorporation of various features such as Riemann SPH, Transport velocity, Wall boundaries, and Thermal diffusion.

**Areas of Improvement:**

1. The work can significantly benefit from a performance analysis with other available open-source SPH solvers, which questions the suitability for HPC applications.
2. Unsucessful evaluation of the original purpose of "Solver-in-the-loop" (SitL), i.e., spatial coarsening, concerns the applicability of methods to other use cases.
3. There has been a significant discussion about using continuous vs discrete adjoints [ma2021, kidger2022, nadarajah2000], which can have significant performance impacts. Evaluating the solvers and implementing the said adjoints could strengthen the technical aspects.

Moreover, I would be interested in more details of what specifics were done to make the solver differentiable, which can serve as a roadmap for other scientists implementing such programs in niche scientific disciplines.

[ma2021] Ma, Yingbo, et al. "A comparison of automatic differentiation and continuous sensitivity analysis for derivatives of differential equation solutions." 2021 IEEE High Performance Extreme Computing Conference (HPEC). IEEE, 2021.

[kidger2022] Kidger, Patrick. "On neural differential equations." arXiv preprint arXiv:2202.02435 (2022).

[nadarajah2000]: Nadarajah, Siva, and Antony Jameson. "A comparison of the continuous and discrete adjoint approach to automatic aerodynamic optimization." 38th Aerospace sciences meeting and exhibit. 2000.

---

### Meta-Review · Area_Chair_QtFq · 2024-03-02

**Recommendation:** Accept (Poster)

**Metareview:**

The paper introduces a differentiable lagrangian fluid simulator based on smooth particle hydrodynamics (SPH), written in JAX. The authors extend the SPH code in LagrangeBench to be differentiable, demonstrating some applications of the said approach. The paper is a useful contribution to the community given the growing interest in SPH codes. There is limited comparisons to other SPH methods but nevertheless it is a well written paper and should be accepted for the poster session.

---

### Decision · Program_Chairs · 2024-03-02

Accept (Poster)